# Text Classification via Large Language Models

**Xiaofei Sun♦∗, Xiaoya Li♣∗, Jiwei Li♦, Fei Wu♦**
**Shangwei Guo♠, Tianwei Zhang♥, Guoyin Wang★**

## Abstract

Despite the remarkable success of large-scale Language Models (LLMs) such as GPT-3, their performances still significantly underperform fine-tuned models in the task of text classification. This is due to (1) the lack of reasoning ability in addressing complex linguistic phenomena (e.g., intensification, contrast, irony etc); (2) limited number of tokens allowed in in-context learning.

In this paper, we introduce **C**lue **A**nd **R**easoning **P**rompting (CARP). CARP adopts a progressive reasoning strategy tailored to addressing the complex linguistic phenomena involved in text classification: CARP first prompts LLMs to find superficial clues (e.g., keywords, tones, semantic relations, references, etc), based on which a diagnostic reasoning process is induced for final decisions. To further address the limited-token issue, CARP uses a fine-tuned model on the supervised dataset for $k$NN demonstration search in the in-context learning, allowing the model to take the advantage of both LLM's generalization ability and the task-specific evidence provided by the full labeled dataset.

Remarkably, CARP yields new SOTA performances on 4 out of 5 widely-used text-classification benchmarks, 97.39 (+1.24) on SST-2, 96.40 (+0.72) on AGNews, 98.78 (+0.25) on R8 and 96.95 (+0.6) on R52, and a performance comparable to SOTA on MR (92.39 v.s. 93.3). More importantly, we find that CARP delivers impressive abilities on low-resource and domain-adaptation setups: using 16 examples per class, CARP achieves comparable performances to supervised models with 1,024 examples per class. Code is available at github.com/ShannonAI/GPT-CLS-CARP [1] [2] [3]

[1] ∗ denotes equal contributions.
[2] ♦Zhejiang University, ♣ Shannon.AI, ★Amazon, ♥Nanyang Technological University, ♠ Chongqing University
[3]xiaoya_li@shannonai.com,{xiaofei_sun, jiwei_li}@zju.edu.cn

## 1 Introduction

Large language models (LLMs) (Radford et al., 2019a; Xue et al., 2020; Zhang et al., 2022a; Rae et al., 2021; Brown et al., 2020; Chowdhery et al., 2022; Ouyang et al., 2022; Thoppilan et al., 2022) have shown the ability for in-context learning (ICL). Given a few demonstration examples, LLMs are prompted to generate results for a new test example, and have achieved performance comparable to supervised baselines or even state-of-the-art results in a variety of natural language processing (NLP) tasks such as question answering (Trivedi et al., 2022), natural language inference, (Schick and Schütze, 2020), named entity recognition (Wang et al., 2023), relation extraction (Wan et al., 2023) and information extraction (Han et al., 2021).

In spite of the success, LLMs with ICL still significantly underperform fine-tuned models for text classification. This is due to two reasons: (1) Text classification requires models with more powerful reasoning abilities to resolve complex linguistic phenomenon including clause composition (e.g., concession, negation, intensification), irony, etc. Recent efforts to improve LLMs' reasoning capabilities (Wei et al., 2022; Kojima et al., 2022; Ye and Durrett, 2022; Zhang et al., 2022b) mainly focus on tackling math problems, and thus are not tailored to addressing the reasoning process necessary for the multitude of intricate linguistic phenomena in text classification; (2) The number of demonstration examples allowed in in-context learning is limited, e.g., the longest context allowed for GPT-3 is 4,096 subtokens. Therefore, LLMs are only able to take the advantage of a small proportion of the training set, performing well below supervised baselines;

In this paper, we introduce **C**lue **A**nd **R**easoning **P**rompting (CARP), an extensible, annotation-free and efficient framework for text classification via large language models. To address the

Figure 1: Examples of **CARP** prompts under zero-shots and few-shot settings. Comparisons of different prompts can be found in Appendix H.

reasoning process necessary for handling the linguistic phenomena in text classification, CARP decomposes the reasoning process into three steps, where LLMs are first prompted to find superficial clues (e.g., keywords, tones, semantic relations, etc) in the given text; next, CARP treats the clues and input as premises and induce a diagnostic reasoning process; and finally determine the final label considering the above two steps. We find this progressive reasoning strategy to be effective in enhancing LLMs' ability in language reasoning involved in text classification. Due to the limited number of tokens allowed in context, a more effective demonstration search is needed. CARP uses a fine-tuned (FT) model on the supervised dataset for $k$NN demonstration search for ICL. Since the fine-tuned model is trained based on task-specific labels, it guarantees that retrieved samples are close to the input sequence with respect to the task. FT-based demonstration search provides a channel to connect LLMs with the full training set, in spite of the limited number of tokens allowed in demonstrations. This strategy lets the model take the advantage of both the LLMs' generalization abilities and all task-specific evidence provided by the training dataset.

Remarkably, CARP yields new SOTA performances on four out of 5 widely-used text-classification benchmarks, 97.39 (+1.24) on SST-2, 96.40 (+0.72) on AGNews, 98.78 (+0.25) on R8 and 96.95 (+0.6) on R52, and a performance comparable to SOTA on MR (92.39 v.s. 93.3). More importantly, we find that CARP delivers impressive ability on low-resource and domain adaptation setups with orders of magnitude fewer training examples. Specifically, CARP achieves comparable performances with 16 examples per class to supervised models trained on the full training set containing more than 1 thousand examples per class. This demonstrates the capabilities of CARP in real-world text classification cases where training data is limited.

## 2 Related Work

### 2.1 Large Language Models

Large language models (LLMs) are models that are trained using self-teaching algorithms on large unlabeled corpora. LLMs can be broadly divided into three categories based on the model architecture. The first category is the encoder-only model like BERT (Devlin et al., 2018). BERT (300M) (Devlin et al., 2018) and its variants (Liu et al., 2019; Sun et al., 2020; Clark et al., 2020; Feng et al., 2020; Sun et al., 2021) adopt the *pre-training then fine-tuning* paradigm for NLP tasks: use masked language models as the main training objective for pretraining, and fine-tune the pretrained model in the annotated downstream datasets. The second category is the decoder-only models like GPT (Radford et al., 2019a). GPT (Radford et al., 2019a) uses the decoder

of an auto-regressive transformer (Vaswani et al., 2017) model for predicting the next token in a sequence. GPT (Radford et al., 2019a) and its variants (Dai et al., 2019; Keskar et al., 2019; Radford et al., 2019b; Chowdhery et al., 2022; Zhang et al., 2022a) also follow the *pre-training then fine-tuning* paradigm. GPT-3 (175B) (Brown et al., 2020) proposes to formalize all NLP tasks as generating textual responses condition on the given prompt. The third category is the encoder-decoder models like T5 (Raffel et al., 2020). T5 (11B) (Raffel et al., 2020) and its variants (Lewis et al., 2019; Xue et al., 2020).

## 2.2 In-context Learning

In-context learning (ICL) generates textual responses (i.e., label words) conditioning on the given prompt (usually) with a few annotated examples for downstream tasks. Li and Liang (2021); Zhong et al. (2021); Qin and Eisner (2021) propose to optimize prompts in the continuous space. Rubin et al. (2021); Das et al. (2021); Liu et al. (2021); Su et al. (2022) introduce different strategies for selecting in-context examples. Lampinen et al. (2022) show that explanations of examples in a few-shot prompt lead to a performance boost. Marasović et al. (2021) find that GPT-3 outperforms other models by a large margin in the explanation generation task. Wei et al. (2022) propose chain-of-thought reasoning and utilized <input, chain-of-thought, output> triples as the prompt for LLMs. Wiegreffe et al. (2021) traine a supervised filter to select explanations generated by GPT-3 on the SNLI and CommonsenseQA tasks.

## 2.3 Text Classification

Text classification is a task that aims to assign predefined labels (e.g., sentiment, topic, etc) to a given text. Earlier work decouple the task into two steps: (1) extract features using neural models such as RNNs (Irsoy and Cardie, 2014; Yang et al., 2016; Wang et al., 2018; Liu et al., 2016; Xie et al., 2020), CNNs (Kim, 2014; Zhang et al., 2015; Lai et al., 2015; Conneau et al., 2016; Wei and Zou, 2019), GCN (Yao et al., 2019), LLMs (Howard and Ruder, 2018; Sun et al., 2019; Chai et al., 2020; Chen et al., 2020; Lin et al., 2021); and (2) feed extracted features into a classifier (Joulin et al., 2016; Tang and Surdeanu, 2023) to obtain the final label.

With LLMs, Schick and Schütze (2020)

reformulate input examples into cloze-style phrases and annotate the unlabeled text. Han et al. (2021) design sub-prompts and applied logic rules to compose sub-prompts into final prompts. Liu et al. (2021) retrieve semantically-similar examples to a test sample to formulate its corresponding prompt. Shi et al. (2022) retrieve label-words-similar examples as demonstrations in prompts.

## 3 Prompt Construction

### 3.1 Overview

We follow the standard prompt-based in-context learning paradigm. Given an input sequence $\boldsymbol{x}_{input} = \{x_1, x_2, ..., x_l\}$, the task of assigning a text-class label to an input text is transformed to generating a pre-defined textual response $\boldsymbol{y} \in \mathcal{Y}_{verb}$ (e.g., positive, negative, etc) conditioning on the prompt $\boldsymbol{x}_{prompt}$ using a language model.

### 3.2 Prompt Construction

The prompt $\boldsymbol{x}_{prompt}$, which is constructed based on $\boldsymbol{x}$, consists of the following three components:

**(1) Task description $\boldsymbol{x}_{desc}$** generally describes the task. For different classification tasks, e..g, sentiment classification, topic classification, etc, descriptions are different. Take the sentiment classification task as an example, the task description is given as follows:
*Classify the overall sentiment of the input as positive or negative*

**(2) Demonstration** consists of a sequence of annotated examples:

$$\{(\boldsymbol{x}_{demo}^1, \boldsymbol{y}_{demo}^1), ..., (\boldsymbol{x}_{demo}^k, \boldsymbol{y}_{demo}^k)\}$$

where $\boldsymbol{x}_{demo}^j, 1 \leq j \leq k$ denotes the $j$th input sequence and $\boldsymbol{y}_{demo}^j$ denotes the text which is transformed from the label, e.g., positive or negative for the binary sentiment classification task. Demonstration serves as two purposes: (1) providing the LLM with evidence to consult on for decision making, which will significantly boost performances; (2) provides an output format that LLM's outputs need to follow, so that the output, which takes the form of natural language, can be further easily transformed to labels. It is worth noting that demonstrations are only needed for the few-shot setup, but not for the zero-shot setup.

**(3) Input** $x_{input}$ is the test text sequence to classify.

The prompt $x_{prompt}$ for a test input is constructed by concatenating the task description $x_{desc}$, a sequence of demonstrations $\{(x_{demo}^1, y_{demo}^1), ..., (x_{demo}^k, y_{demo}^k)\}$, and the test sequence $x_{test}$, which can be given as follows:

$$\{x_{desc}; \backslash n; <\text{demo}>^1; \backslash n; ...; <\text{demo}>^k; \backslash n; x_{test}\}$$

## 3.3 Demonstration Sampling

The few-shot setup requires demonstrations sampled from the training set. Strategies that we explore include:

**Random Sampling** a straightforward strategy from samplings is to randomly sample $k$ examples from the training set $\mathcal{D}_{train}$ for a text sequence $x_{test}$.

$k$**NN Sampling** The key disadvantage for random sampling is that there is no guarantee that selected samples are semantically related to the input sequence. One straightforward alternative is to sample examples that are similar to the test sequence using $k$NN search (Khandelwal et al., 2020). In this process, the test sequence $x_{test}$ is first mapped to a vector $v_{test}$ using an encoder model $f$. Then using $v_{test}$ as the query, we search through the entire training set $\mathcal{D}_{train}$ to retrieve $k$ nearest text sequence to get $k$ nearest data examples $\mathcal{N} = \{x_j, y_j\}_{j=1}^k$ as demonstrations. We use the following encoder models to obtain sentence representations and similarity scores:

**SimCSE** is a contrastive learning model for sentence embeddings(Gao et al., 2021). [4]

**Finetuned Model** FT for short. The key disadvantage of SimCSE (Gao et al., 2021) and other general semantic encoding models (Reimers and Gurevych, 2019; Seonwoo et al., 2022; Sun et al., 2022) is that it measures the general semantic similarity but is not specifically tailored to the text classification task. To resolve this issue, CARP uses the model fine-tuned on the training dataset as the $k$NN encoder model. Specifically, we first fine-tune a Roberta model on the training data. Next we use the [CLS] embedding as the sentence level representation for KNN search. Since the fine-tuned model is trained based on task-specific labels, it guarantees that retrieved samples are close to the

input sequence with respect to the task. Using fine-tuned model provides a channel to connect LLMs with the full training set, in spite of the limited number of tokens allowed in demonstrations. This strategy lets the model take the advantage of both the LLMs' generalization abilities and all task-specific evidence provided by the training dataset.

## 4 Clues Collecting and Reasoning

To enhance the models' reasoning ability in addressing linguistic phenomenon tailored to text classification, we propose a progressive reasoning strategy that involves clue collection, reasoning and decision making. This process also mimics how human decisions: where we first collect evidence from the input, separating chaff from wheat; next we piece together local evidence to form a global picture, which leads to final decision making. Next we first given an overview of the the clue collecting and reasoning process, and then describe implementation details.

### 4.1 Overview

**Collecting Clues** For a test sequence, clues are local fact evidence such as keywords, phrases, contextual information, semantic meaning, semantic relationships, tones, references, etc. The following is an example for clues of an input:
**Input**: *Steers turns in a snappy screenplay that curls at the edges; so clever you want to hate it.*
**Clues**: *"snappy", "clever", "want to hate it" are clues for determining the sentiment of the input sentence.*

**Reasoning** For reasoning, the LLM is prompted to go beyond superficial keywords to mine deeper perspectives, considering language phenomenon such as negation, intensification, irony, etc), and piece together local evidence to form the final decision. The following example shows the reasoning process to decide the sentiment of the above example based on the evidence collected:
*1. The phrase "snappy screenplay" implies that the screenplay is of a high quality and is well-crafted.*
*2. The phrase "curls at the edges" implies that the screenplay is cleverly written.*
*3. ...*

**Decision Making** Based on the reasoning process, the model makes the decision for the sentiment of the given input:

---

[4]We use `Sup-SimCSE-RoBERTa-Large` model as an encoder model.

*Overall, the clues and reasoning process point to a positive sentiment for the input sentence.*

The merits for the incorporation of clue finding and reasonings are as follows: (1) it prompts the model to progressively think and make decisions: clue finding focuses more on superficial features such as keywords, while reasoning makes deeper justifications based on superficial features; (2) clue finding and reasoning serve as a tunnel to let human intervene: in the few-shot setup, where clues and reasons need to be prepared in advance for demonstrations, we can modify them as we see fit. This is extremely helpful for trouble shooting in the prompt-construction stage for error corrections; (3) from an interpretation and uncertainty estimation perspective, clues and reasoning in few-shot setups are human-readable influence functions;

### 4.1.1 zero-shot scenario

In the zero-shot setup, as no demonstration is allowed, no concrete example for clues and reasons can be provided in the prompt. In this way, we only add requests asking the model to output clues and reasons in the prompt.

### 4.1.2 few-shot scenario

In the few-shot setup , we need to prepare clues and reasonings for all examples in the training set in advance as all training examples have chances to be selected as demonstrations given different test inputs. Previous efforts in math problems (Wei et al., 2022; Kojima et al., 2022; Ye and Durrett, 2022; Zhang et al., 2022b) prepare hand-drafted reasoning for a few examples, and always use these example as demonstrations. This strategy does not fit for our situation as it is extremely time-intensive to manually generate clues and reasonings for all training examples. To resolve this issue, we harness LLMs for automatic clue and reasoning generation, where we ask LLMs to generate clues and reasoning based on both the input and its corresponding label.

**Clue Generation**   For a given training example *<text>* paired with the label word *<label-word>* (e.g., positive), we ask LLM to generate clues that indicate the label:

> *List CLUES (i.e., keywords, phrases, contextual information, semantic meaning, semantic relationships, tones, references) that support the sentiment determination of the input (limit to 15 words).*

*INPUT: <text>*
*SENTIMENT: <label-word>*

**Reasoning Generation** Based on clues generated clues, the input, and the label, we ask LLMs to generate reasoning details[5]:

> *Based on the input and clues, articulate the diagnostic reasoning process that supports the sentiment determination of the input.*
> *INPUT: <text>*
> *LABEL: <label-word>*
> *CLUES: <clues>*
> *REASONING:*

Given the generated clues and reasonings for all training examples, at test time, when K-nearest examples are selected demonstrations, its corresponding clues and reasons are concatenated to the demonstration. In this way, each demonstration example is composed by a `(text, clues, reasons, golden label word)` pair. Examples for prompts with clues and reasons are shown in Figure 4. In this way, for a test example, by following the format of demonstrations, the LLM will first output clues, then reasons, and at last decisions.

## 4.2 Voting

Unlike conventional discriminative models for text classification, which generate deterministic results during inferences, LLMs for in-context learning are generative models and generate distinct textual responses with diverse sampling strategies in multiple runs. We consider the following voting strategies in the paper:

- **Majority Vote**: the final result is the most frequent prediction among multiple runs.
- **Weighted Probability Vote**: the final result is the one with weighted summed probability from multiple runs.

## 5   Experiments

In order to evaluate the effectiveness of the proposed method, we conduct experiments on two setups: (1) full training setup, where the model has the access to the full training data; and (2) low-resource setup, where the model can only access partial training dataset. The low-resource setup

---

[5]LLMs often generate long responses, in order to ensemble more demonstrations in prompts, we use *"limit to 50 words"*. After conducting an analysis of the generated responses, we find that LLMs can explain the reason within limited words.

| | SST-2 | AGNews | R8 | R52 | MR | Average |
|---|---|---|---|---|---|---|
| **Supervised Methods** | | | | | | |
| RoBERTa-Large (Liu et al., 2019) | 95.99 | 95.55 | 97.76 | 96.42 | 91.16 | 95.38 |
| DeBERTa (He et al., 2020) | 94.75 | 95.32 | 98.33 | 96.32 | 90.19 | 94.99 |
| RoBERTa-GCN (Lin et al., 2021) | 95.80 | **95.68*** | 98.2 | 96.1 | 89.7 | 95.10 |
| XLNet (Yang et al., 2019) | **96.10*** | 95.55 | - | - | - | - |
| VLAWE (Ionescu and Butnaru, 2019) | - | - | - | - | **93.3*** | - |
| GCN-SB (Zeng et al., 2022) | - | - | **98.53*** | **96.35*** | 87.59 | - |
| **Zero-shot Setting** | | | | | | |
| Vanilla (Brown et al., 2020) | 91.55 | 90.72 | 90.19 | 89.06 | 88.69 | 90.04 |
| CoT (Kojima et al., 2022) | 92.11 | 91.25 | 90.48 | 91.24 | 89.37 | 90.89 |
| **CARP** | 93.01 | 92.60 | 91.75 | 91.80 | 89.94 | 91.82 |
| **Few-shot Setting ($k$=16)** | | | | | | |
| *Random Sampler* | | | | | | |
| Vanilla (Brown et al., 2020) | 92.36 | 91.74 | 91.58 | 91.56 | 89.15 | 91.28 |
| CoT (Kojima et al., 2022) | 94.56 | 95.02 | 92.49 | 92.03 | 89.91 | 92.80 |
| **CARP** | 96.20 | 95.18 | 97.60 | 96.19 | 90.03 | 95.04 |
| *SimCSE kNN-Sampler* | | | | | | |
| Vanilla (Brown et al., 2020) | 93.90 | 93.50 | 94.36 | 92.40 | 89.59 | 94.05 |
| CoT (Kojima et al., 2022) | 94.21 | 94.28 | 95.07 | 92.98 | 90.27 | 93.69 |
| **CARP** | 95.69 | 95.25 | 97.83 | 96.27 | 90.74 | 95.16 |
| *FT kNN-Sampler* | | | | | | |
| Vanilla (Brown et al., 2020) | 94.01 | 94.14 | 95.57 | 95.79 | 90.90 | 94.08 |
| CoT (Kojima et al., 2022) | 95.48 | 94.89 | 95.59 | 95.89 | 90.17 | 94.40 |
| **CARP** | 96.80 | 95.99 | 98.29 | 96.82 | 91.90 | 95.97 |
| **CARP** (WP Vote) | 97.39 | 96.40 | 98.78 | 96.95 | 92.39 | 96.38 |

Table 1: Accuracy performances of different settings on benchmarks. We report mean results over 5 runs. The GPT-3 denotes `text-davinci-003`. In few-shot experiments, we sample 16 annotated examples ($k$=16) for every test instance. **\*** indicates existing SOTA results. "WP Vote" denotes weighted probability vote.

better mimics real-world situations where training data is limited. For the full training setup, we follow the standard train/dev/test split. For the low-resource setup, we randomly sample $n$ instances per class ($n$ in $\{16, 128, 256, 512, 1024\}$) from the benchmark training set. The sampled subset forms a new training set to test different models' abilities in the low-resource situations. During experiments, we train models/demonstrations with the new training set.

We conduct experiments on five widely-used datasets, including SST-2 (Socher et al., 2013), R8, R52[6], AGNews (Zhang et al., 2015) and Movie Review (MR) (Pang and Lee, 2005). More details of the benchmarks and low-resource datasets can be found in Appendix D.

For zero-shot and few-shot experiments, we use InstructGPT-3 (Ouyang et al., 2022) (`text-davinci-003`, 175B) as the backbone. Due to the input token limitation, we use $k = 16$ for few-shot setups. Prompts on the five datasets are shown in Appendix H. Model hyper-parameters can be found in Table 13 [7]. We use **Vanilla** to denote the conventional ICL approach where LLMs are prompted to generate labels, use **CoT** (Kojima et al., 2022) to denote the baseline

---

[6]R8 and R52 are original from https://www.cs.umb.edu/~smimarog/textmining/datasets/

[7]During experiments, we find that CARP is robust with different hyper-parameters. Experimental results can be found in Appendix G.3

| Dataset | Model | $n$=16 | $n$=128 | $n$=256 | $n$=512 | $n$=1024 |
|---|---|---|---|---|---|---|
| **SST-2** | FT RoBERTa | 51.52 | 52.31 | 53.89 | 70.49 | 90.30 |
| | GPT-3 Vanilla | 90.15 | 90.36 | 91.70 | 93.86 | 94.68 |
| | GPT-3 CoT | 89.66 | 90.19 | 90.80 | 94.42 | 94.89 |
| | GPT-3 CRAP | 90.48 | 91.07 | 91.77 | 94.03 | 95.20 |
| **AGNews** | FT RoBERTa | 21.87 | 38.19 | 40.08 | 50.18 | 78.09 |
| | GPT-3 Vanilla | 89.47 | 89.63 | 90.54 | 93.02 | 94.79 |
| | GPT-3 Zero-shot-CoT | 89.66 | 90.16 | 91.70 | 94.86 | 95.28 |
| | GPT-3 CRAP | 90.16 | 90.94 | 91.07 | 94.08 | 95.48 |
| **R8** | FT RoBERTa | 11.29 | 48.19 | 60.18 | 70.70 | 88.68 |
| | GPT-3 Vanilla | 89.15 | 90.27 | 91.70 | 94.00 | 94.91 |
| | GPT-3 CoT | 90.49 | 90.88 | 91.81 | 95.42 | 95.75 |
| | GPT-3 CRAP | 90.23 | 91.03 | 91.77 | 95.56 | 96.67 |
| **R52** | FT RoBERTa | 38.29 | 39.10 | 59.18 | 67.19 | 81.53 |
| | GPT-3 Vanilla | 89.15 | 90.04 | 90.29 | 91.88 | 92.06 |
| | GPT-3 CoT | 89.46 | 90.02 | 90.73 | 93.20 | 94.12 |
| | GPT-3 CRAP | 90.82 | 91.00 | 95.85 | 94.36 | 96.27 |
| **MR** | FT RoBERTa | 51.20 | 52.11 | 53.58 | 68.29 | 88.37 |
| | GPT-3 Vanilla | 86.04 | 88.68 | 88.99 | 89.80 | 90.18 |
| | GPT-3 CoT | 86.26 | 89.00 | 90.01 | 90.16 | 90.89 |
| | GPT-3 CRAP | 86.54 | 87.19 | 89.63 | 90.01 | 91.20 |

Table 2: Experimental results on low-resource ($n$ example per class) settings. We compare fine-tuned RoBERTa-Large with 16-shots GPT-3 setting. For GPT-3, we use SimCSE (Gao et al., 2021) to retrieve 16 annotated examples from the low-resource train set.

| | FT RoBERTa on SST-2 Train | FT RoBERTa on Yelp Train |
|---|---|---|
| SST-2 Test | 95.99 | 88.78 |
| Yelp Test | 92.38 | 96.04 |
| | **CARP with SST-2 demon.** | **CARP with Yelp demon.** |
| SST-2 Test | 96.80 | 96.29 |
| Yelp Test | 95.94 | 96.32 |

Table 3: Results for Yelp test set when using in-domain/out-of-domain $k$NN sampler and demonstrations source. We use FT $k$NN Sampler to retrieve demonstrations on the corresponding train set.

that mimics the chain-of-thought strategy and **CARP** to denote the proposed method.

## 5.1 Models for Comparison

**Supervised models** are trained on the trained set naturally constitute baselines to compare with. We use the six models, including RoBERTa-Large, RoBERTa-GCN, DeBERTa, XLNet, GCN-SB, and VLAWE. Details of the models and hyper-parameters are shown in Appendix G.2:

**Few-shot Setup** For demonstration sample strategies in the few-shot setup, we consider the following strategies for comparison: (more details can be found in Section 3.3):

- **Random Sampler**: randomly samples $k$ examples.
- **SimCSE $k$NN-Sampler**: samples $k$NN based on SimCSE.
- **FT $k$NN-Sampler**: sample $k$NN using **F**ine-**T**uned RoBERTa representations.

## 5.2 Results on the full training set

Experimental results are shown in Table 1. As can be seen, performances of few-shot setups consistently outperform zero-shot setups. In terms of sampling strategies in the few-shot setups, we observe that simcse KNN-sampler outperform random sampler, illustrating the importance of adding demonstrations that are relevant to the test input in the few-shot setup. We also observe that FT KNN-sampler consistently outperforms simcse KNN-sampler. This shows that, the fine-tuned model, which takes the advantage of the full training set, serves as a better retriever for task-specific demonstration retrieval than the general-purposed SimCSE retriever.

For different reasoning strategies, we first observe that the CoT strategy outperforms the vanilla strategy, which straightforwardly asks LLMs to generate results without further reasoning steps. CARP consistently outperforms CoT across all benchmarks, i.e., +1.48, +0.97, +2.76, + 3.29, +0.47 respectively on SST-2, AGNews, R8, R52 and MR datasets. This demonstrates the necessity of building models with complex linguistic phenomena involved in text classification, and the effectiveness of CARP in doing this job.

Compared with supervised learning baselines, we find that the vanilla model using LLM underperforms supervised baselines, while few-shot CoT is able to obtain slightly worse or comparable results agains supervised baselines. Notably, single CARP outperforms fine-tuned RoBBERTa on all benchmarks. Using WP voting strategies, CARP yields new SOTA performances on four out of the 5 datasets, 97.39 on SST-2 (+1.24), 96.40 (+0.72) on AGNews, 98.78 (+0.25) on R8 and 96.95 (+0.6) on R52, and a performance comparable to SOTA on MR (92.39 v.s. 93.3).

## 5.3 Results on low-resource settings

To estimate low-resource circumstances, we sample $n = \{16, 128, 256, 512, 1024\}$ instances for each class as low-resource setups. Experimental results are shown in Table 2. As can be seen, when the training set size is extremely small (i.e., 16 or 128 sentences), and the performance of the supervised model is far below CARP. Even with only 16 examples to train on, the accuracy of CARP of SST-2 already around 90%, whereas supervised models' performance is similar to random guess. This demonstrates the strong generalization ability

of CARP in the low-resource setup. As we anticipated, the $k$NN search efficiency improved at a faster rate as the amount of the training data increases; Enlarging the training dataset increases the chances that the chosen examples will correspond to the input, resulting in improved results. Specifically, using 16 examples per class, CARP achieves comparable performances to supervised models with 1,024 examples per class; using 512 instance per class annotation data, CARP achieves comparable performances to supervised models trained on the full set.

## 5.4 Domain Adaptation

It is unclear whether training models on the specific dataset for retrieving demonstrations is essential. In this subsection, we conduct an analysis on using demonstrations from out-of-distribution datasets.

We use SST-2 and Yelp, and the task is to determine the positive or negative polarity of the given text. SST-2 and Yelp are from different domains: SST-2 are snippets from Rotten Tomatoes[8], whereas Yelp consists of product reviews from the online website. Experimental results are shown in Table 3. SST-2 train & SST-2 test means that demonstrations are from the SST-2 dataset and test is performed on SST-2 dataset; Yelp train & SST-2 test means demonstrations are from Yelp and test is performed on SST-2 dataset. We see a significant decrease (-7.2%, 95.99% v.s.88.78% ) in performance when switching SST-2 train to Yelp train using supervised RoBERTa, which illustrates that supervised models are very sensitive to the out-of-distribution data. On the contrary, we only observe a slight decrease in performance (-0.5%, 96.80% v.s. 96.29%) when switching SST-2 train to Yelp-2 train on SST-2 test, illustration the greater capabilities of CARP on the domain adaptation situations. This means CARP is very robust when training and test are not from the same domain.

## 6 Ablation Studies

### 6.1 Impact of the number of demonstrations

We explore the effect of the number of demonstrations in prompts using SST-2 . Results for CARP using different sampling strategies are shown in Figure 2. As can be seen, performances improve as the number of demonstrations increases, which is in line with our expectation.

---

[8] https://www.rottentomatoes.com/

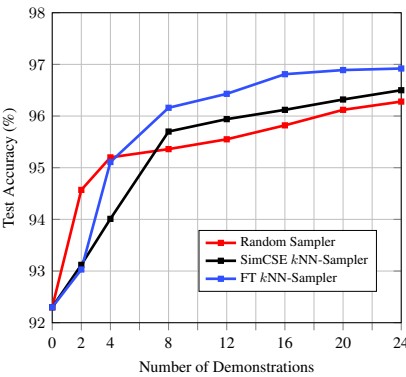

Figure 2: Performances v.s. the number of demonstrations for CARP.

| Prompts | SST-2 | R8 |
|---|---|---|
| CARP | 96.80 | 98.29 |
| w/o Text | 92.28 | 94.18 |
| w/o Clue | 95.48 | 95.29 |
| w/o Reason | 95.72 | 97.82 |
| w/o Label | 96.53 | 98.18 |

Table 4: The effect of components on the SST-2 dataset with different strategies.

## 6.2 The effect of components in demonstrations

CARP uses `(text, clues, reasons, golden label word)` pairs as demonstrations. In this subsection, we exploit the influence of each component in `(text, clues, reasons, golden label word)` by removing it from prompts. Experimental results are shown in Table 4. As shown in Table 4, text in demonstrations has the greatest impact, followed by clue, reason and label.

## 6.3 The effect of different types of label words

Label words denote words generated by LLMs that indicate the label of the input. We explore the impact of using different kinds of label words:

- **Position index**: e.g., one, two, three, etc.
- **Annotation words**: e.g., positive, negative. [9]
- **Synonyms words**: e.g., great, terrible.
- **Flipped words**: words that are contrary to original target meanings. e.g., "positive" to denote the negative polarity, "negative" to denote the positive polarity.
- **Random words**: randomly choose words in the vocabulary.

---

[9]GPT-3 generates the same label words for binary sentiment classification task.

| Strategy | Label Words(+,-) | CARP |
|---|---|---|
| Position Index | One, Two | 95.66 |
| Annotation Words | Positive, Negative | **96.86** |
| Synonyms Words | Great, Terrible | 96.27 |
| Flipped Words | Negative, Positive | 64.63 |
| Random Words | Cf, Ng | 95.06 |
| Special Tokens | <POS>, <NEG> | 96.65 |

Table 5: Label words and results on the SST-2 dataset with different strategies. "+" represents positive polarity; "-" denotes negative polarity.

| Ranking | SimCSE | FT |
|---|---|---|
| | CARP | |
| Random | 95.39 | 95.99 |
| High-to-Low | 95.22 | 96.71 |
| Low-to-High | 96.39 | 96.80 |

Table 6: Accuracy scores on SST-2 when assembling demonstrations with different ranking strategies.

- **Special tokens**: tokens that do not have semantic meaning. They are independent of the input and added for a certain purpose. e.g., <cls>, <mask>.

Results are shown in Table 5. As can be seen, using annotation words as label words achieves the best performances. We also observe a significant performance decrease when flipped words are used as label words in demonstrations.

## 6.4 The effect of demonstration order

During experiments, we find that the ranking order of demonstration affect final results. In this subsection, we further investigate the influence of orders of demonstrations. Orders the demonstrations we investigate include:

- **Random**: randomly shuffle retrieved demonstrations.
- **Low-to-High**: demonstrations with lower similarity scores come first. Therefore demonstrations with higher similarity scores are placed closer to the test sequence, which is placed at the end of the prompt.
- **High-to-Low**: demonstrations with lower similarity scores are placed closer to the test sequence.

As shown in Table 6, the performance is sensitive the ordering of the demonstrations. The low-to-high ordering achieves the best performance compared to the random and high-to-low ordering.

# 7 Conclusion

In this paper, we introduce **C**lue **A**nd **R**easoning **P**rompting (CARP) for text classification task. CARP yields new SOTA performances on 4 out of 5 widely-used text-classification benchmarks. More importantly, we find that CARP delivers impressive abilities on low-resource and domain-adaption setups. In the future, we would like to explore CARP on more natural language understanding tasks.

## Acknowledgements

This work is supported by the National Key R&D Program of China (SQ2022AAA010214).

## Limitations

Despite the overall promising results, CARP still faces the following shortcomings: (1) clues that are contributing for making decisions are hard to annotate; (2) LLMs still suffer from the token limit issue.

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

## A Prompts

In Figure 3, we present examples of **vanilla**, **CoT** (Kojima et al., 2022) and the proposed **CARP** prompts in the zero-shot setting.

In Figure 4, we present examples of **vanilla**, **CoT** (Kojima et al., 2022) and the proposed **CARP** prompts in the few-shot ($k$=1) setting.

## B Experimental results

In Table 7, we present the experimental results on text classification subsets.

## C More ablation studies

### C.1 The influence of clues

Cclues includare keywords, phrases, contextual information, semantic meaning, semantic relationships, tones, references that support making decisions. We remove different types of words in clues and evaluate its influence on SST-2 and R8 datasets. Editing prompts achieve this goal. The original prompt for clue collecting is *List CLUES (i.e., keywords, phrases, contextual information, semantic meaning, semantic relationships, tones, references) that support the sentiment determination of the input.* If we want to remove *keywords & phrases*, we just remove them from the prompt.

- **w/o keywords & phrases**: keywords and phrases are surface evidence for making decisions such as *"like"*, *"hate"*.
- **w/o contextual information & semantic meaning**: contextual information and semantic meaning are meaning in sentences/paragraphs such as *The author express his happiness*.
- **w/o semantic relationships**: semantic relationships refer to relations between subjects such as *"emotional danger" suggests a romantic and thrilling relationship between Idemoto and Kim that creates a positive sentiment.*.
- **w/o tones**: tones are the general mood of the text such as *The sentence is expressed in an objective tone*.
- **w/o references**: references are mentions of commonsense facts or books such as *The reference to the popular, comedic character "Ferris Bueller" implies that the kid is seen in a positive light.*.

Experimental results are shown in Table 8. For R8 and SST-2 datasets, keywords play the key role for GPT predictions.

### C.2 Quality of the reasoning process

In this paper, we use LLMs to generate rationable explanations instead of human editing. Therefore, the quality of generated reasoning process affects the final results. In this subsection, we sample 500 training *(text, clues, reason, label)* pairs and evaluate the generated reasoning process from the following perspectives:

**(1) Reliability:** Inspired by the emergent generalization ability of LLMs, we use zero-shot GPT-3 (175B) as the self-critique model to evaluate the quality of generated reasoning processes. To be specific, we ask the GPT-3 to return yes/no if the generated reasoning process supports making decisions for the input text. If the GPT-3 returns "yes", it denotes that the reasoning process is reliable for making decisions. If the GPT-3 returns "no", it represents that the reasoning process is not reliable.

The prompt for SST-2 is shown as follows:
*Is the following REASONING process supporting determinate sentiment label to INPUT? Please answer Yes or No.*
*INPUT: <text>*
*REASONING: <reasoning-process>*
where *<text>* is the text sequence for the data and *<reasoning-process>* is generated reasoning process.

**(2) Fluency:** use LLMs to generate reasoning explanations is a reference-free text generation task. We use perplexity to evaluate the generated text.

**(3) Logic Faithful:** previous work often use models, which are trained on natural language

**This is an overall sentiment classifier for movie reviews. Classify the overall SENTIMENT of the INPUT as Positive or Negative.**

INPUT: press the delete key

SENTIMENT: Neutral ✗

**(a) Vanilla Zero-shot**

**This is an overall sentiment classifier for movie reviews. Classify the overall SENTIMENT of the INPUT as Positive or Negative.**

INPUT: press the delete key

SENTIMENT: Let's think step-by-step. The input does not contain any words that would indicate a sentiment, so it is not possible to classify the sentiment as either positive or negative. ✗

**(b) CoT Zero-shot**

**This is an overall sentiment classifier for movie reviews.**
**First, list CLUES (i.e., keywords, phrases, contextual information, semantic relations, semantic meaning, tones, references) that support the sentiment determination of input..**
**Second, deduce the diagnostic REASONING process from premises (i.e., clues, input) that supports the INPUT sentiment determination (Limit the number of words to 130).**
**Third, based on clues, reasoning and input, determine the overall SENTIMENT of INPUT as Positive or Negative.**

INPUT: press the delete key

CLUES: delete key
REASONING: The phrase "delete key" implies an action of removing something, which could be interpreted as a negative sentiment.
SENTIMENT: Negative ✓

**(c) CARP Zero-shot**

Figure 3: Examples of zero-shot prompting methods for the text classification task: **(a)** represents for the **vanilla** prompting method; **(b)** denotes for the **Chain-of-Thought (CoT)** (Kojima et al., 2022) prompting method; **c** represents for the proposed **CARP** prompting method.

inference datasets, to determine whether the given "hypothesis" logically follows from the "premise". However, lacking annotation datasets, NLI-trained models can not generalize across multiple domains (e.g., opinion, reviews, news). Since then, we use 16-shot ICL with GPT-3 to evaluate whether the generated rationable explanations can be entailed from the input text. If the InstructGPT responds with "entailment", it denotes that the generated reasoning process is logic faithful with the text. Otherwise, it represents the reasoning process is not faithful to the text. We sample training instances from the SNLI dataset (Bowman et al., 2015) as demonstrations. And prompts are shown as follows:

*Given the premise and hypothesis, please justify whether the HYPOTHESIS can be entailed from the PREMISE. Please return yes or no.*
*PREMISE: <text>*
*HYPOTHESIS: <reasoning-process>*

Evaluation results are shown in Table 9. As can be seen, the reliability percentages for SST-2 and R5 are higher than 95%. This indicates that it is feasible to use the model-generated reasoning process as part of the prompts to augment ICL performances. The perplexity of generated reasoning text is smaller than 4, which denotes that the generated reasoning text is fluent. And scores of logic faithful are larger than 93%, which is in line with our expectation that LLMs can generate reasonable explanations.

# D Dataset

SST-2 (Socher et al., 2013), R8, R52[10], AGNews (Zhang et al., 2015) and MR (Movie Review) (Pang and Lee, 2005).

- **SST-2**: The original data in SST-2 are sampled from snippets of Rotten Tomatoes HTML files. We use the same train/dev/test splits with Socher et al. (2013).
- **R8 and R52**: R8 and R5211 are two subsections of the Reuters collection, containing 8 and 52 classifications, respectively. The R8 dataset is composed of 5,485 documents for training and 2,189 documents for testing. The R52 dataset is composed of 6,532 training and 2,568 test documents.
- **AGNews**: The AG News consists of news articles from the AG's corpus. The dataset contains 30,000 training and 1,900 testing examples for each class.
- **MR (Movie Review)**: The MR contains reviews of films for determining whether a sentiment is either positive or negative. The corpus has 10,662 reviews. We follow (Tang et al., 2015) and use the same train/test split.

**This is an overall sentiment classifier for movie reviews. Classify the overall SENTIMENT of the INPUT as Positive or Negative.**

**INPUT: noyce films it more as a shocking history lesson than as drama.**
**SENTIMENT: Positive**

**INPUT: press the delete key**
**SENTIMENT: Neutral** ✗

**(a) Vanilla One-shot**

**This is an overall sentiment classifier for movie reviews. Classify the overall SENTIMENT of the INPUT as Positive or Negative.**

**INPUT: noyce films it more as a shocking history lesson than as drama.**
**SENTIMENT: Let's think step-by-step. The input states that the films are "more" something than something else. This implies that the films are at least somewhat of the first thing mentioned. In this case, the films are more of a "shocking history lesson" than "drama". This implies that the sentiment is positive, as the films are more of a history lesson than drama. The overall sentiment is Positive.**

**INPUT: press the delete key**
**SENTIMENT: Let's think step-by-step. The input does not provide any context or opinion, so it is impossible to determine the sentiment. Therefore, the overall sentiment is Neutral.** ✗

**(b) CoT One-shot**

**This is an overall sentiment classifier for movie reviews.**
**First, present CLUES (i.e., keywords, phrases, contextual information, semantic relations, semantic meaning, tones, references) that support the sentiment determination of input.**
**Second, deduce a diagnostic REASONING process from premises (i.e., clues, input) that supports the sentiment determination (Limit the number of words to 130).**
**Third, determine the overall SENTIMENT of INPUT as Positive or Negative considering CLUES, the REASONING process and the INPUT.**

**INPUT: noyce films it more as a shocking history lesson than as drama.**
**CLUES: - Positive: "more," "shocking," "history lesson," "drama." - Negative: None.**
**REASONING: The input contains positive words such as "more," "shocking," and "drama," which suggest that the sentiment is positive. Additionally, the phrase "history lesson" implies that the film is educational and informative, which is generally seen as a positive thing. Therefore, the sentiment of the input is positive.**
**SENTIMENT: Positive**

**INPUT: press the delete key**
**CLUES: - Positive: None. - Negative: "delete key."**
**REASONING: The input contains the phrase "delete key," which implies that something is being removed or erased. This is generally seen as a negative action, suggesting that the sentiment of the input is negative.**
**SENTIMENT: Negative** ✓

**(c) CARP One-shot**

Figure 4: Examples of few-shot (*k*=1) prompting methods for the text classification task: **(a)** represents for the **vanilla** prompting method; **(b)** denotes for the **Chain-of-Thought (CoT)** (Kojima et al., 2022) prompting method; **(c)** represents for the proposed **CARP** prompting method.

# E  Case Analysis

We show generated *clues, reasoning explanations* examples in Table 12.

# F  Baselines

As mentioned in Section 5.1, we use the following supervised models as baselines. More details of the models are as follows:

- **RoBERTa-Large**:We fine-tune RoBERTa-Large (Liu et al., 2019) on the training set.
- **RoBERTa-GCN**:Lin et al. (2021) constructs heterogeneous graph networks on top of the RoBERTa-Large (Liu et al., 2019) model.
- **DeBERTa**:He et al. (2020) improve RoBERTa by using disentangled attention mechanism and an enhanced mask decoder.
- **XLNet**:Yang et al. (2019) propose a generalized autoregressive pretraining method that enables learning bidirectional contexts.

- **GCN-SB**:Zeng et al. (2022) propose a simplified boosting algorithm, which makes CNN learn the samples misclassified by GCN again.
- **VLAWE**:Ionescu and Butnaru (2019) obtain document embeddings based on aggregating the differences between each codeword vector and each word vector (from the document) associated to the respective codeword.

# G  Hyper-parameters

## G.1  GPT API Hyper-parameters

Hyper-parameters for GPT-3 are shown in Table 13.

## G.2  Fine-tuning Hyper-parameters

We fine-tune RoBERTa and RoBERT-GCN on 4 NVIDIA 3090 GPUs with FP16. Model hyper-parameters are tuned on the validation set, where learning rate {2e-5, 3e-5, 4e-5}, batch size {16, 32, 32}, a dropout rate of 0.3, a weight decay of 0.01, a warmup proportion of 0.01.

---

[10]R8 and R52 are from https://www.cs.umb.edu/~smimarog/textmining/datasets/

| | SST-2 | AGNews | R8 | R52 | MR | Average |
|---|---|---|---|---|---|---|
| **Supervised Methods** | | | | | | |
| RoBERTa-Large | 95.99 | 95.55 | 97.76 | 96.42 | 91.16 | 95.38 |
| RoBERTa-GCN | 95.80 | 95.68 | 98.2 | 96.1 | 89.7 | 95.10 |
| **Zero-shot Setting** | | | | | | |
| Vanilla | 91.55 | 90.72 | 90.19 | 89.06 | 88.69 | 90.04 |
| Zero-shot-CoT | 92.11 | 91.25 | 90.48 | 91.24 | 89.37 | 90.89 |
| **CARP** | 94.41 | 93.18 | 93.29 | 92.69 | 90.03 | 92.72 |
| **Few-shot Setting** | | | | | | |
| *Random Sampler* | | | | | | |
| Vanilla | 91.36 | 91.48 | 90.60 | 90.68 | 89.15 | 90.65 |
| Zero-shot-CoT | 92.56 | 92.65 | 92.49 | 92.03 | 89.91 | 91.93 |
| **CARP** | 94.41 | 93.18 | 93.29 | 92.69 | 90.03 | 92.72 |
| *SimCSE kNN-Sampler* | | | | | | |
| Vanilla | 93.90 | 93.50 | 94.36 | 92.40 | 89.59 | 92.75 |
| Zero-shot-CoT | 94.21 | 94.28 | 95.07 | 92.98 | 90.27 | 93.36 |
| **CARP** | 95.99 | 95.53 | 95.31 | 93.84 | 90.64 | 94.26 |
| *FT kNN-Sampler* | | | | | | |
| Vanilla | 94.01 | 94.14 | 95.57 | 95.79 | 90.90 | 94.08 |
| Zero-shot-CoT | 95.48 | 94.89 | 95.59 | 95.89 | 90.17 | 94.40 |
| **CARP** | 96.62 | 95.97 | 98.13 | 96.12 | 91.86 | 95.74 |

Table 7: Accuracy performances of different settings on test subsets (results are over 5 runs). GPT-3 denotes `text-davinci-003`. In few-shot experiments, we sample 16 annotated examples ($k$=16) per prompt. "MJ Vote" is short for majority vote. "WP Vote" denotes weighted probability vote.

| Prompts | SST-2 | R8 |
|---|---|---|
| Clues | 96.80 | 98.29 |
| w/o keyword&phrase | 96.21 | 96.91 |
| w/o contextual info. | 96.23 | 97.10 |
| w/o semantic relations | 96.30 | 97.38 |
| w/o tones | 96.40 | 97.35 |
| w/o reference | 96.50 | 97.19 |

Table 8: Label words and results on the SST-2 dataset with different strategies.

| | Reliability(%) ↑ | Fluency(ppl) ↓ | Logic Faithful(%) ↑ |
|---|---|---|---|
| SST-2 | 96.18 | 3.89 | 95.20 |
| R8 | 95.34 | 3.29 | 94.55 |

Table 9: Results for evaluating the quality of generated reasoning explanation. We sample 500 (text, reason) instances for SST-2 and R8.

| Dataset | Task | # Label | Source | # Train | # Dev | # Test |
|---|---|---|---|---|---|---|
| SST-2 | sentiment | 2 | review | 6,920 | 872 | 1,821 |
| AGNews | topic | 4 | news | 96,000 | 24,000 | 7,600 |
| R8 | topic | 8 | news | 4,941 | 544 | 2,189 |
| R52 | topic | 52 | news | 5,905 | 627 | 2,568 |
| MR | sentiment | 2 | reviews | 6,398 | 710 | 3,554 |

Table 10: Benchmark Dataset

| Dataset | Task | # Label | Source | # Train | # Dev | # Subtest |
|---|---|---|---|---|---|---|
| SST-2 | sentiment | 2 | review | 6,920 | 872 | 728 |
| AGNews | topic | 4 | news | 96,000 | 24,000 | 760 |
| R8 | topic | 8 | news | 4,941 | 544 | 875 |
| R52 | topic | 52 | news | 5,905 | 627 | 1,027 |
| MR | sentiment | 2 | reviews | 6,398 | 710 | 888 |

Table 11: Dataset Subsets

## G.3 The influence of hyper-parameters

We investigate the effect of model hyper-parameters including temperature, frequency penalty. We conduct experiments with Instruct-GPT3 on the SST-2 dataset.

**Temperature** The temperature $\tau$ controls the generated text variety when another hyper-parameter $top_p$=1. More higher $\tau$, more variety is introduced. When $\tau$ is close to 0, the model generates the same result with the greedy decoding method. To exploit the effect of temperature $\tau$, we set $\tau$ from 0 to 1.0. Experimental results are shown in Table G.3. We tokenize the response text with GPT-Tokenizer[11] and then count the number of tokens.

| $\tau$ | SST-2 Accuracy |
|---|---|
| $\tau = 0.0$ | 96.39 |
| $\tau = 0.2$ | 96.48 |
| $\tau = 0.4$ | 96.40 |
| $\tau = 0.6$ | 96.59 |
| $\tau = 0.8$ | 96.68 |
| $\tau = 1.0$ | 96.70 |

## H Prompts

---

[11] https://platform.openai.com/tokenizer

| INPUT & GOLD LABEL | CLUES | REASONING |
|---|---|---|
| **INPUT**: johnnie to and wai ka fai are sure to find an enthusiastic audience among american action adventure buffs, but the film 's interests may be too narrow to attract crossover viewers
**GOLD LABEL**: Negative | - Positive Clues: enthusiastic, action, adventure.
- Negative Clues: narrow, crossover | The diagnostic reasoning process supporting the sentiment determination of the input is that the phrase "johnnie to and wai ka fai are sure to find an enthusiastic audience among american action adventure buffs" suggests that the film is likely to be well-received by this specific demographic. However, the subsequent phrase "but the film's interests may be too narrow to attract crossover viewers" implies that the film is unlikely to appeal to a broader audience, suggesting a negative sentiment. Therefore, the overall sentiment of the input is negative. |
| **INPUT**: moore sell globe unit moore resources inc said agreed principle sell globe inc unit cyprus minerals co undisclosed terms sale expected close mid april subject certain conditions including board approval companies globe produces silicon metals products used chemical aluminum iron rubber semiconductor industries reuter
**GOLD LABEL**: Acquisitions | - Acquisition (agreed, sell, unit, sale, close, board approval)
- Companies (Moore Resources Inc, Globe Inc, Cyprus Minerals Co)
- Products (silicon, metals, chemical, aluminum, iron, rubber, semiconductor)
- Reuter (news agency) | The input mentions an acquisition agreement between Moore Resources Inc and Globe Inc, and the sale is expected to close in mid-April, suggesting an Acquisitions topic. The input also mentions Cyprus Minerals Co, silicon and metals products which are used in chemical, aluminum, iron, rubber, and semiconductor industries, and a Reuter news agency, all of which support the Acquisitions topic. |

Table 12: Examples of generated clues and reasoning for demonstrations.

| Parameter | Value |
|---|---|
| Engine Name | text-davinci-003 |
| Max Tokens | 200 |
| Temperature | 0.7 |
| Top P | 1 |
| Frequency Penalty | 0.0 |
| Presence Penalty | 0.0 |
| Best Of | 1 |

Table 13: OpenAI API Hyper-parameters.