# OpenReview forum: "Text Classification via Large Language Models"
_EMNLP/2023/Conference — EMNLP 2023 Findings_

### Official Review · Reviewer_jdhr · 2023-07-29

**Soundness:** 4

**Excitement:**

4: Strong: This paper deepens the understanding of some phenomenon or lowers the barriers to an existing research direction.

**Paper Topic And Main Contributions:**

This paper proposes an approach to document-level sentiment classification task based on LLMs and in-context learning, using prompting with examples which enables reasoning to get correct labels.
The prompt asks the model to generate clue keywords related to positive or negative sentiments, and reasoning process that support the decision of labels, in addition to the final answer.  Also examples of (input, clues, reasoning and label) are added to the prompt for one-short and few-shot settings which can be automatically generated from the subset of training data.
The experimental results show the SotA scores in 4 out of 5 tasks, and they outperformed the supervised fine-tuned results.  Also this paper provides extensive discussion including ablation studies, alternative labels and consideration of low-resource scenario.

**Questions For The Authors:**

A. The ablation study shows that it works even without the gold label in the example.  How does the examples help without gold labels?


**Reasons To Accept:**

* Good design of prompt including examples.  The difference from the Chain-of-Thought method is clear.
* The experimental results show the SotA scores in 4 out of 5 tasks, and they outperformed the supervised fine-tuned results.
* The proposed method addresses the low-resource settings and domain adaptability.
* The discussion of alternative labels is interesting.


**Reasons To Reject:**

* The notion of in-context learning has been already proposed.
* The document-level classification task is too popular and simple and its application is limited.

**Reproducibility:**

4: Could mostly reproduce the results, but there may be some variation because of sample variance or minor variations in their interpretation of the protocol or method.

**Reviewer Confidence:**

3: Pretty sure, but there's a chance I missed something. Although I have a good feel for this area in general, I did not carefully check the paper's details, e.g., the math, experimental design, or novelty.

**Typos Grammar Style And Presentation Improvements:**

* Figure 2 should be placed on the top of the column.
* Numbers in Table 1 and 2 are too small.  Since they are key results they should be displayed with larger letters.
* CRAP->CARP in Table 2

---

> ### Author Rebuttal · Authors · 2023-08-29
>
> Thanks for your very encouraging and insightful comments.
>
> **Q1**: The ablation study shows that it works even without the gold label in the example. How does the examples help without gold labels?
>
> **A1**: Thank you for asking. We use SST-2 as an example to illustrate.
> As mentioned in Section 4, each demonstration in CARP consists of four parts: text, CLUE, reasoning, and gold label. In Section 6.2, after removing the gold label, the CLUE and reasoning in the demonstration still provide evidence that helps to determine overall sentiment. Therefore, they can still help GPT to determine the overall sentiment of the test input. An example of CARP (1-shot) without gold label is as follows:
>
>
>
> > **<CARP-Task-Description>**
> **INPUT**: noyce films it more as a shocking history lesson than as drama.
> **CLUES**: - Positive: "more," "shocking," "history lesson," "drama." - Negative: None.
> **REASONING**: The input contains positive words such as "more," "shocking," and "drama," which suggest that the sentiment is **positive**. Additionally, the phrase "history lesson" implies that the film is educational and informative, which is generally seen as a **positive** thing.
>   **INPUT**: press the delete key.
>
> **Q2**: Regarding “Typos Grammar Style And Presentation Improvements”
>
> **A2**: Thank you for kind reminder. We will correct typos and enlarge experimental tables in the updated version.

---

### Official Review · Reviewer_9Y4M · 2023-08-02

**Soundness:** 4

**Excitement:**

3: Ambivalent: It has merits (e.g., it reports state-of-the-art results, the idea is nice), but there are key weaknesses (e.g., it describes incremental work), and it can significantly benefit from another round of revision. However, I won't object to accepting it if my co-reviewers champion it.

**Missing References:**

Extract feature and feed into classifier:
Zheng Tang, Mihai Surdeanu; It Takes Two Flints to Make a Fire: Multitask Learning of Neural Relation and Explanation Classifiers. Computational Linguistics 2023; 49 (1): 117–156. doi: https://doi.org/10.1162/coli_a_00463

**Paper Topic And Main Contributions:**

This paper proposes a framework to use LLMs for text classification task. It first guide the LLMs to extract the key clues from the input and use the clues to get the final decision. To improve the few-shot performance, authors also propose to train kNN to generate the examples.

**Questions For The Authors:**

Did you evaluate this method on some open-source LLMs, like Llama?

**Reasons To Accept:**

The paper is overall written very well and is easy to follow.
Experiments are thorough and complete. Using multiple methods, multiple tasks and ablation studies lead the generalizability of the paper's claims.

**Reasons To Reject:**

The authors should not claim their framework as "annotation-free" while using a kNN trained on supervised dataset.

**Reproducibility:**

4: Could mostly reproduce the results, but there may be some variation because of sample variance or minor variations in their interpretation of the protocol or method.

**Reviewer Confidence:**

4: Quite sure. I tried to check the important points carefully. It's unlikely, though conceivable, that I missed something that should affect my ratings.

---

> ### Author Rebuttal · Authors · 2023-08-29
>
> We want to thank you for the very encouraging comments.
>
> **Q1**: Regarding the missing reference and the "annotation-free" claim.
>
> **A1**: We will add the missing reference and fix the "annotation-free claim" in the next version.
>
> **Q2**: Did you evaluate this method on some open-source LLMs, like Llama?
>
> **A2**: Thank you for asking. It is a great idea to try CARP on open-source LLMs. We will run the experiments and update the results in the next version.

---

### Official Review · Reviewer_uxYB · 2023-08-04

**Soundness:** 2

**Excitement:**

2: Mediocre: This paper makes marginal contributions (vs non-contemporaneous work), so I would rather not see it in the conference.

**Paper Topic And Main Contributions:**

This paper discusses tasks related to text classification through large models. The author first points out two reasons why large language models (LLMs) perform poorly in text classification tasks: a lack of strong reasoning capability and the performance limitations caused by window length restrictions. Subsequently, the author proposes the Clue And Reasoning Prompting (CARP) method. This method instructs large models on how to use superficial clues to enhance their reasoning ability during demonstrations. The most effective demonstrations are retrieved using kNN demonstration search. The author conducted experiments on five text classification datasets using text-devinci-003, and the results showed that this method performed well.

Contributions：
1. The author outlined the reasons why large models underperform in text classification, and introduced CARP to enhance their performance.
2. Experiments were conducted on five commonly used text classification datasets.

**Questions For The Authors:**

1. The experiment chose only one large model, which is text-devinci-003. Considering that ChatGPT has been released for eight months, why wasn't it considered in the experiment?

2. What do you think is the essential difference between your method and CoT?

**Reasons To Accept:**

1. The writing is good and the method is easy to follow.

**Reasons To Reject:**

1. The author mentions that large models perform poorly due to inadequate reasoning capability and performance constraints caused by window length limitations. However, the experimental results show that large models perform well (with an average accuracy of 90% under zero-shot) which makes it confusing.
2. There is insufficient analysis and experimental support for the two reasons given. At the same time, there is no obvious answer to whether the strong enough reasoning capability proposed by the author is needed in these selected text classification datasets.
3. There is a lack of novelty, and clues reasoning can be considered a natural extension of cot. Knn sampling is also something that has already been proposed in existing work.
4. The method proposed by the author has limited improvement in performance.

**Reproducibility:**

4: Could mostly reproduce the results, but there may be some variation because of sample variance or minor variations in their interpretation of the protocol or method.

**Reviewer Confidence:**

4: Quite sure. I tried to check the important points carefully. It's unlikely, though conceivable, that I missed something that should affect my ratings.

---

> ### Author Rebuttal · Authors · 2023-08-29
>
> Thanks for your feedback. We will answer your question in order below.
>
> **Q1**: The experiment chose only one large model, which is text-devinci-003. Considering that ChatGPT has been released for eight months, why wasn't it considered in the experiment?
> **A1**: ChatGPT's API was open-access on March 2, 2023, just two months before the EMNLP submission deadline (June 16). At that time, we had already completed all the experiments reported in the paper. Given the constrained timeframe, it was not feasible to reconduct all experiments using ChatGPT. However, we did undertake experiments with ChatGPT on five text classification datasets and plan to report the results in the next version. To answer Q1, we show experimental results (16-shot) of ChatGPT on SST-2 and R8 as follows:
>
> | Dataset             | SST-2 | R8 |
> | ------------------- | :---: | :----: |
> | RoBERTa-Large      | 95.99 | 97.76  |
> | Few-shot (FT-Retriever, k=16) |   |   |
> | Vanilla     | 94.6  | 94.0 |
> | CoT     | 95.7 | 94.8 |
> | CARP    | **96.9** | **95.8** |
>
> **Q2**: What do you think is the essential difference between your method and CoT?
> **A2**: We would like to clarify the difference between CoT ("Let’s think step-by-step.”) and the proposed CARP. The primary distinction between the proposed CARP and CoT lies in the problem domain and methodology. CoT emphasizes a step-by-step approach, typically for mathematical problems. In contrast, CARP is designed specifically for text classification, addressing the complexities of linguistic elements such as clause composition (including concession, negation, and intensification), irony, and more. Thus, CARP is better suited for comprehensively handling the nuanced linguistic challenges in text classification.
>
> **Q3**: There is insufficient analysis and experimental support for the two reasons given. At the same time, there is no obvious answer to whether the strong enough reasoning capability proposed by the author is needed in these selected text classification datasets.
> **A3**: We conducted a quality analysis of the generated reasoning. Due to the page limit, we present the analysis in the appendix. Please refer to Appendix C.2 (line 979- line 1042). We will move this section to the main content in the next version.
>
> **Q4**: The method proposed by the author has limited improvement in performance.
> **A4**: Performance improvements are remarkable. As mentioned in Section 5, CARP yields new state-of-the-art (SOTA) performances on 4 out of 5 widely-used text-classification benchmarks.
>
> **Q5**: The author mentions that large models perform poorly due to inadequate reasoning capability and performance constraints caused by window length limitations. However, the experimental results show that large models perform well (with an average accuracy of 90% under zero-shot), which makes it confusing.
> **A5**: Large language models perform poorly on the text classification task. We will use SST-2 as an example to illustrate. A Bi-LSTM neural network with random initialization achieves 87.5 accuracy [1] on SST-2, while text-davinci-003 achieves 91.5. Considering the number of parameters of LLMs and data scales used during training LLMs, the improvement of +3.9 under zero-shot is limited.
>
> [1] Socher et al., Recursive Deep Models for Semantic Compositionality Over a Sentiment Treebank. EMNLP 2013.

---

### Meta-Review · Area_Chair_qST3 · 2023-09-18

**Recommendation:** 3

**Metareview:**

This paper explores text classification in GPT-3, and introduces a method (CARP) to help models use superficial clues for better classification performance. The paper reports new SOTA performances in 4 of 5 chosen text-classification benchmarks.

The reviewers agree that the paper is clear and well-written, generally find the methods and experiments solid, and appreciate the demonstrated performance improvement.

The reviewers also raise concerns about the strength of the motivations and impact of the paper, pointing out that baseline model performance is already high on these tasks, and the improvements are fairly incremental. The reviewers would also prefer to see stronger support for the claims made about the reasons for poor model performance in these domains, (alongside stronger motivation and justification for the claim that this performance needs improvement in the first place).

---

### Decision · Program_Chairs · 2023-10-07

**Decision:**

Accept-Findings

**Comment:**

This paper explores text classification in GPT-3, and introduces a method (CARP) to help models use superficial clues for better classification performance. The paper reports new SOTA performances in 4 of 5 chosen text-classification benchmarks.

The reviewers agree that the paper is clear and well-written, generally find the methods and experiments solid, and appreciate the demonstrated performance improvement.

The reviewers also raise concerns about the strength of the motivations and impact of the paper, pointing out that baseline model performance is already high on these tasks, and the improvements are fairly incremental. The reviewers would also prefer to see stronger support for the claims made about the reasons for poor model performance in these domains, (alongside stronger motivation and justification for the claim that this performance needs improvement in the first place).